

# Efficient retention of mud drives land building on the Mississippi Delta plain

Christopher R. Esposito[1], Zhixiong Shen[2‡], Torbjörn E. Törnqvist[1], Jonathan Marshak[1, †], Christopher White[1]

[1]Department of Earth and Environmental Sciences, Tulane University, 6823 St. Charles Avenue, New Orleans, Louisiana 70118-5698, USA.
[2]Department of Marine Science, Coastal Carolina University, PO Box 261954, Conway, South Carolina 29528, USA.

*Correspondence to*: Christopher R. Esposito (cresposito@gmail.com)

‡ Christopher R. Esposito and Zhixiong Shen contributed equally to this work, and should be considered co-first authors

† Current Address: Department of Geological Sciences, California State Polytechnic University, 3801 West Temple Avenue, Pomona, California 91768, USA



**Abstract.** Many of the world's deltas – home to major population centers – are rapidly degrading due to reduced sediment supply, making these systems less resilient to increasing rates of relative sea-level rise. The Mississippi Delta faces some of the highest rates of wetland loss in the world. As a result, multi-billion dollar plans for coastal restoration by means of river diversions are currently nearing implementation. River diversions aim to bring sediment back to the presently sediment-starved

5    delta plain. Within this context, sediment retention efficiency (*SRE*) is a critical parameter because it dictates the effectiveness of river diversions. Several recent studies have focused on land building along the open coast, showing *SRE*s ranging from 5 to 30%. Here we measure the *SRE* of a large relict crevasse splay in an inland, vegetated setting that serves as an appropriate model for river diversions. By comparing the mass fraction of sand in the splay deposit to the estimated sand fraction that entered it during its life cycle we find that this mud-dominated sediment body has an *SRE* of ≥75%, i.e., dramatically higher

10   than its counterparts on the open coast. Our results show that transport pathways for mud are critical for delta evolution and that *SRE* is highly variable across a delta. We conclude that sediment diversions located in settings that are currently still vegetated are likely to be the most effective in reversing land loss and providing long-term sustainability.





# 1 Introduction

Most large rivers do not transport sufficient sediment to the coast to fill the accommodation that will be created on their delta plains due to rapid 21st century sea-level rise (Giosan et al., 2014). This shortfall ensures a global retreat of deltaic coasts and

presents an existential threat to some of the densest human populations, most valuable economic infrastructure, and most vibrant ecologies on Earth (Ericson et al., 2006; Giosan et al., 2014). To mitigate land loss, sediments can be distributed to vulnerable or otherwise important locations with controlled diversions of sediment-laden river water (Day et al., 2007; Kim et al., 2009; Paola et al., 2011; Giosan et al., 2013; Smith et al., 2015; Auerbach et al., 2015; Coastal Protection and Restoration Authority of Louisiana, 2017). The most effective techniques for such diversions are subject to debate (Blum and Roberts,

2009; Kim et al., 2009; Nittrouer and Viparelli, 2014a; Blum and Roberts, 2014; Nittrouer and Viparelli, 2014b), but it is clear that maximizing sediment retention efficiency (*SRE*) is a critical concern (Blum and Roberts, 2009; Paola et al., 2011). Because fine sediments are highly mobile in suspension and delta plains are often regarded as inefficient traps for mud (Giosan et al., 2014), much of the literature on diversions has focused on extracting sandy material from the trunk channel (Nittrouer et al., 2012a; Nittrouer and Viparelli, 2014b; Meselhe et al., 2016). However, mud comprises 80% or more of the incoming sediment

load in most rivers (Giosan et al., 2014) and often dominates their delta-plain deposits.

Published estimates of *SRE* (Nittrouer et al., 1995; Allison et al., 1998; Goodbred and Kuehl, 1998; Törnqvist et al., 2007; Blum and Roberts, 2009; Day et al., 2016; Roberts et al., 2016) vary from 5 to 80%, a range that is too wide to be useful for planning purposes, but which suggests that the specific depositional setting is an important control. For example, the Wax Lake Delta (WLD), a well-studied lobe of the Mississippi Delta (MD) (Figure 1a) that serves as a key natural analog for

diversions, is sand-dominated (Roberts et al., 2003; Kim et al., 2009) and has been estimated to have an *SRE* ranging from 5 to 30% ( Törnqvist et al., 2007; Kim et al., 2008; Roberts et al., 2016). Here we propose that vegetated inland settings that are protected from marine processes on the open coast can be highly efficient in trapping sediment, especially mud. We test this hypothesis by quantifying the *SRE* of a large crevasse splay in the MD, and demonstrate the essential role that mud plays in aggrading delta plains. Our work highlights the contrast in depositional style between a protected inland setting and locations

on the open coast.

We set this study in the Attakapas Crevasse Splay (ACS), a ~60 km$^2$ landform that was constructed from 1.2 to 0.6 ka (Shen et al., 2015) and initially discharged into a mature cypress swamp that is now preserved as a regionally continuous wood peat bed underlying overbank strata (Törnqvist et al., 2008) in the Lafourche subdelta of the MD (Figure 1). We choose a crevasse splay because such features are ubiquitous along all major distributary channels and are thus important building blocks of the

MD (Davis, 1993; Day et al., 2007; Shen et al., 2015); because of the well-established stratigraphy based on 132 cores and a chronology based on extensive $^{14}$C and OSL dating in the region (Törnqvist et al., 1996; Shen et al., 2015); and because its



size and timescale of activity are in line with those of planned diversion projects in the MD (Coastal Protection and Restoration Authority of Louisiana, 2017).

## 2 Measuring sediment retention efficiency

There are only a handful of studies that have attempted to tie the bulk sedimentary properties of a recent deposit to sediment-
transport properties in the river that created it (Törnqvist et al., 2007; Kim et al., 2009; Giosan et al., 2013; Day et al., 2016), and we are unaware of any with a subsurface data set as rich as the one available for the ACS. While many workers have published on the "river side" issues concerning diversions, including the sediment available (Kesel, 1988; Blum and Roberts, 2009; Allison et al., 2012) and the physics of extracting sediment from the trunk channel (Allison and Meselhe, 2010; Meselhe et al., 2012; Nittrouer et al., 2012a; Allison et al., 2013), only recently have researchers begun to investigate "basin side" issues
that impact *SRE* (Xu et al., 2016). The availability of detailed sediment-transport data from the modern Lower Mississippi River (LMR) (Allison et al., 2012) provides a unique opportunity to connect fluvial sediment budgets to the sediments preserved in the delta.

We estimate the *SRE* of the ACS with a 3D model (Figure 2) based on the 132 cores augmented by 53 grain-size analyses. These data are used to quantify the sand fraction (>62.5 µm) in the ACS deposit ($S_d$), and we estimate the sand fraction in the
ACS input ($S_i$) from published hydraulic and sediment-load data from the modern LMR. If we compare the two sand fractions, and stipulate that 100% of the sand that enters the system is retained, *SRE* is obtained with Eq. (1):

$$SRE = \frac{S_i}{S_d} \qquad (1)$$

This scheme measures the loss of mud (<62.5 µm) from the splay. When $S_d$ is very close to $S_i$, *SRE* approaches 100%. Therefore, a deposit with a grain-size composition that closely resembles its input is an efficient sediment trap. In the MD, fed by a river whose current sediment load is ~80% mud, this means that sand-rich deposits are the products of ineffective sediment traps.

## 2.1 Sediment texture

All 132 cores were described in the field at 10 cm increments following the United States Department of Agriculture (USDA) texture classification system (cf. Shen et al., 2015). Texture classes encountered were very fine sand (vfS), sandy loam (SL), silt loam (SiL), silty clay loam (SiCl), silty clay (SiC), and clay (C). Organic-rich clays are denoted as humic clay (HC). The sand fraction for each texture class was determined by grain-size analysis of 53 samples taken from three separate cores (Table 1). For our analysis we combined SL and vfS into a single class to which we applied the sand fraction measured for vfS
samples. This is consistent with our objective to estimate an upper limit of sand content in the deposit. We combined SiC, HC,



and C into a single class as well. Samples were treated with hydrochloric acid to remove carbonates and with hydrogen peroxide to remove organic matter, then wet sieved through a 106 µm screen. Both fractions were dried and weighed, then analyzed for grain size. The coarse fraction was analyzed with a Retsch Technology CAMSIZER and the fine fraction with a Horiba LA-300 Laser Particle Size Analyzer. The resulting sediment distributions were weighted and patched together to obtain a continuous curve that was used to determine the sand fraction.

## 2.2 Crevasse splay 3D model and deposit sand fraction

All 3D modeling was done by means of Schlumberger's Petrel geomodeling software. The basal bounding surface of the ACS was generated with R, using the gstat package (Pebesma, 2004). The ACS consists of primarily muddy facies that overlie a wood peat bed that predates occupation of the region by the precursor of the modern LMR, Bayou Lafourche (Törnqvist et al., 1996). The earliest Lafourche deposits feature a 1-3 m thick clay bed that transitions abruptly into a silty matrix with sandy ribbons embedded within it (Figure 3). We interpret the clay to silt transition as the base of the ACS, and the coarser sandy deposits as splay channel deposits.

The top bounding surface of the splay is the modern land surface as measured by LiDAR. The picked subsurface elevation of the clay to silt transition was linearly regressed on the elevation of the local land surface ($R^2 = 0.58$). This regression yields a useful result because higher elevations are correlated with thicker deposits, which have differentially compacted the underlying strata (Törnqvist et al., 2008). We use the regression function to generate a surface for the transition, which we refine by kriging the residuals and adding the result back to the surface. Adding the kriged residuals ensures that the regression surface matches our core picks exactly, and exploits local excursions from the regression trend to improve the estimate based on nearby data points. Wherever possible, the edges of the ACS were taken to be the intersection between the top and basal surfaces. When the surfaces diverge due to the occurrence of neighbouring splays, the edge was chosen visually based on the local topography.

Channel bodies in the ACS are identifiable both as narrow alluvial ridges and as coarser sediment bodies in the subsurface. All channel bodies were modeled to extend through the full thickness of the splay. This choice makes the channel bodies appear somewhat less sandy than they might be in reality, but also makes them larger; the end result is a slightly sandier estimate of $S_d$ overall. The supplemental spreadsheet includes a sensitivity analysis on the effects of varying the ratio of channel to non-channel sediment volumes.

We determined the average sand fraction for the channel and non-channel portions of the ACS separately (Table 1). A volume-weighted average of these sand fractions yields the estimate of overall $S_d$ for the ACS.



### 2.3 Input sand fraction

We estimated the yearly averaged sand fraction input into a 5 m deep crevasse channel emanating from a 30 m deep trunk channel. We obtained the depth of the trunk channel, Bayou Lafourche, from previous investigations of the region (Fisk, 1952). Shen et al. (2015) showed that the ACS deposit has a total thickness of up to 10 m, that it was active during two episodes of

rapid aggradation, and that similar thicknesses accumulated near the inlet during each episode. In keeping with these data, we used 5 m as a representative estimate of the depth of the primary crevasse splay feeder channel during its lifetime. This value is similar to that of other well-studied crevasse systems in the MD (Farrell, 1987). The flow depth of the trunk channel was assumed not to vary significantly, consistent with measurements in the modern LMR where at 100 km from the shoreline the flow depth varies less than 2 m throughout the year (Nittrouer et al., 2012b).

To estimate the input sand fraction ($S_i$), we used modern suspended sediment measurements from the United States Geological Survey (USGS) gauge station at Belle Chasse, Louisiana ("BC" in Figure 1a) and published estimates of the sediment composition and hydraulic properties in the modern LMR (Nittrouer et al., 2012b; Ramirez and Allison, 2013). We divided river discharge into five bins, each of which was treated separately (Table 2) to estimate the channel width integrated concentration of sand and mud in suspension in the uppermost 5 m of the 30 m deep trunk channel. We then used rating

curves (Allison et al., in review) to compute the discharge of suspended sediments corresponding to each daily water discharge. The lowest flow bin is assumed to carry mud but no sand, consistent with modern observations (Nittrouer et al., 2008).

To obtain the sand fraction in the uppermost 5 m of the water column we used 1) a vertical profile of relative suspended sediment concentration for mud and sand in each discharge bin, 2) the total suspended load for each sediment class and

discharge bin (estimated from gauge measurements at Belle Chasse), and 3) log-law velocity profiles for each discharge bin. All calculations can be seen in the '$S_i$' tab of the supplemental spreadsheet. Sand concentration is assumed to follow a Rouse profile (Rouse, 1936), while mud is well mixed. We use shear velocities in the range of published modern LMR measurements (Nittrouer et al., 2012b; Ramirez and Allison, 2013) when calculating Rouse and velocity profiles.

We assumed that the sand load obtained from Belle Chasse was composed of 125 µm and 250 µm grains, in a 35/65 split,

consistent with measurements in the modern LMR (Ramirez and Allison, 2013). We calculated Rouse profiles for each sand fraction separately, and then added them together in proportion to obtain the concentration profile for suspended sand.

We adapted methods used in recent work in the modern LMR (Nittrouer et al., 2011) and the Rouse equation to calculate profiles of relative concentrations of suspended sand. The equation for relative concentration of suspended sand is:

$$\frac{r_n}{1-r_n} = [\left(\frac{H-Z}{Z}\right) - (\frac{z_a}{H-z_a})]^p \qquad (2)$$





where $r_n$ represents the relative concentration of sand in suspension in depth layer $n$, and is a function of channel depth ($H$), height above the bed ($Z$), thickness of the bedload transport layer ($z_a$, which here is on the order of $10^{-4}$ m), and the Rouse number $p = \frac{w_s}{\kappa u_*}$. To calculate the Rouse number we use the Dietrich method (Dietrich, 1982) to determine sediment fall velocity ($w_s$), and take literature values for shear velocity ($u_*$) (Nittrouer et al., 2012b; Ramirez and Allison, 2013) and von Karman's constant $\kappa$.

Our Rouse profiles are not calculated using a near-bed reference concentration, and thus they describe the shape of the sand concentration profile but not its magnitude. We use these curves to define a set of proportionality constants $k_n$, that relate the $r_n$ values to the bottom value, $r_1$.

$$r_n = k_n r_1 \qquad (3)$$

We then note that

$$Q_s = \sum v_n c_n \qquad (4)$$

where $Q_s$ is the channel integrated suspended sediment load for the size class in question, $v_n$ is the velocity in vertical layer $n$, and $c_n$ is the width-integrated sediment concentration in the same layer. For simplicity, we show the case where each layer is of unit thickness and is the full width of the channel. The width of the channel will affect the magnitude of the concentration values, but will not affect the suspended sand fraction, which is the objective of the computation.

Using the proportionality constants defined in Eq. (3) we now expand Eq. (4) and obtain a value for $c_1$.

$$Q_s = v_1(k_1 c_1) + v_2(k_2 c_1) + \cdots + v_n(k_n c_1) \qquad (5)$$

$$c_1 = \frac{Q_s}{\sum_1^N v_n k_b} \qquad (6)$$

The remaining $c_n$ values can be obtained by multiplying $c_1$ by the appropriate $k_n$.

Note that we have calculated the width-integrated sand concentration in our depth interval rather than the sediment load. We do this because crevasse splays can distort local flow fields, but concentrations in a turbulent flow can be inherited from upstream. To calculate the sand fraction entering the ACS we assume that the splay takes a constant fraction of the trunk



channel's discharge throughout the hydrograph, which we call $\gamma$. This assumption finds support in data from the West Bay, Baptiste Collette, and Grand Pass crevasses in the birdfoot delta of the modern LMR (Allison et al., 2012).

The sand fraction delivered to the ACS, $S_i$, is then calculated as

$$S_i = \frac{\sum_j c_{sand,j}\gamma Q_{w,j}}{\sum_j c_{sand,j}\gamma Q_{w,j} + \sum_j c_{mud,j}\gamma Q_{w,j}} \tag{7}$$

where $c_{sand,j}$ and $c_{mud,j}$ are average concentrations in the uppermost 5 meters for discharge bin $j$, and $Q_{w,j}$ is the water discharge in the trunk channel at each bin integrated through the entire Belle Chasse record. We make no assumptions about $\gamma$ other than that it is nonzero. The value of $\gamma$ does not affect the result of this calculation.

## 3 Results

The 3D model (Figure 2) shows a half-lens shaped deposit with a maximum thickness of ~10 m where flow entered from Bayou Lafourche. The total volume of the ACS is $1.62\times10^8$ m$^3$. Former channels can be identified as topographic ridges (Figure 1b) that correspond with sandier ribbons in the subsurface (Figure 3). While the narrow alluvial ridges associated with crevasse channels are the most striking topographic features on the splay, channel deposits constitute only 15.6% of the ACS deposit. Sandy textures (sandy loam, very fine sand) make up 17% of the channel bodies, but only 11% of the non-channel deposits. On the scale of the entire ACS, silty textures (silt loam, silty clay loam) are by far the most common.

Texture descriptions were calibrated to sand fraction by grain-size analysis of the 53 samples (Table 1). The results of this analysis show that the fraction of sediment mass in the ACS that is represented by sand-sized grains is 0.045±0.017. We use the mean value as our estimate of $S_d$ when calculating $SRE$.

Using modern suspended sediment data we estimate that the yearly averaged input sand fraction ($S_i$) is 0.066 (Table 2).

Combined with the estimated ACS sand fraction ($S_d$) of 0.045 (Table 1), we obtain a $SRE$ value that exceeds 100%. This value is the result of the historical decline in sediment load documented in the modern LMR which is treated more fully below.

## 4 Discussion

### 4.1 Uncertainty in estimating $SRE$

The ACS boundary is defined by the extent of silt-dominated deposits and therefore includes all sandy sediment bodies encased within it. Any reasonable splay boundary will exclude the most mobile sediments, so we proceed with the understanding that some fine sediments were lost to the surrounding environment. For example, the bank of Lake Verret near



the downstream end of the ACS is convex where splay-derived sediments have partially filled it (Figure 1b), suggesting a loss of fine sediments across our defined boundary.

Unlike $S_d$, we are unable to measure $S_i$ directly, so we apply modern sediment-load measurements to the prehistoric channel geometry to estimate the sand fraction that was discharged to the ACS. The Lafourche channel had a depth similar to the

modern LMR channel downstream of Belle Chasse (Fisk, 1952), which gives us confidence to apply modern hydraulic parameters in our analysis. We do not consider the effect of channel planform on the cross-channel distribution of sediments.

Spillways that are part of the modern flood-control system prevent the largest floods, which had magnitudes substantially greater than what is allowed today (Barry, 1998). The discharges in our largest bin (>30,000 m$^3$ s$^{-1}$) carry only 3% of the average yearly suspended sediment load, implying that these large magnitude events probably occurred infrequently enough

that their impact on the crevasse sediment budget was small. The omission of larger but even less frequent floods would have even less of an effect. To the extent that this omission is important it means that the value for $S_i$, and therefore *SRE*, is underestimated.

Perhaps the largest uncertainty is the sand and mud fraction carried by the LMR prior to human modifications. It is well documented that the suspended sediment load of the modern LMR was dramatically reduced by dams in the mid-20[th] century

(Kesel, 1988), but because it is likely that pre-dam sediment loads were elevated due to widespread agricultural activity in the drainage basin (Keown et al., 1986; Tweel and Turner, 2012), it is difficult to estimate the suspended sediment loads that prevailed from 1.2 to 0.6 ka. Recent modeling results suggest that the suspended sand load delivered to the MD is buffered from upstream change for timescales on the order of 1000 years (Nittrouer and Viparelli, 2014a). With that in mind, the events that may have substantially changed the suspended sand load are either so recent (dams in the tributaries, rapid deforestation

associated with expanding agriculture; <200 years ago) or so long past (glacial outwash floods; >10,000 years ago (Rittenour et al., 2007) that it is reasonable to apply modern sand loads to the Lafourche channel.

The question of paleo-mud loads is more challenging, as there is no accepted estimate for the magnitude of the sediment load increase due to expanding agriculture. Instead of choosing one particular mud load (and implied sand fraction) we performed our $S_i$ and *SRE* calculations for a range of possibilities. Assuming that the modern mud load reflects a 30% reduction from the

prehistoric condition leads to an *SRE* value of approximately 100%, thereby imposing a lower limit on mud load reduction. A 50% reduction in mud load, which is consistent with the limited data that are available for the LMR prior to widespread intensive agriculture (Tweel and Turner, 2012), corresponds to an *SRE* of 75%. We therefore estimate that the ACS had an *SRE* value between 75 and 100%, which corresponds to a 30 to 50% reduction in suspended mud load. It is important to note that the conservative assumptions that underpin our estimates of $S_i$ and $S_d$ make 75% a conservative lower bound on *SRE*.





The LMR discharge was shared between Bayou Lafourche and the modern LMR when the ACS was active (cf. Törnqvist et al., 1996). Since the partitioning of the discharge between these two distributaries is unknown, we assume a roughly equal partition of the sand and mud load.

## 4.2 Implications for delta evolution and sustainability

The ACS is overwhelmingly composed of mud, with sand-sized grains accounting for only ~5% of its mass. Such a low sand fraction indicates a system that is highly efficient in retaining mud. Accretion rates of $1-4$ cm yr$^{-1}$ persisted in the ACS for centuries (Shen et al., 2015). Such rates, if attained today, may be sufficient to keep up with present-day rates of relative sea-level rise in the MD ($1.3 \pm 0.9$ cm yr$^{-1}$) (Jankowski et al., in revision) after accounting for enhanced compaction due to sediment loading (Törnqvist et al., 2008). The ACS is one of the largest splays in the region, and with its well-developed channel network

it likely drew more water, from greater depths, than other crevasse splays along Bayou Lafourche. It is therefore likely to be comparatively enriched in sand. Because of this, and because some fine material must have been lost beyond our downstream boundary, our estimate of $S_d$ likely resides near the upper limit for crevasse splays in the MD.

The 5% sand fraction found in the ACS stands in contrast to the much higher values observed in prograding coastal delta lobes such as the WLD, where the sand fraction might be as high as 50%. A value of 67% sand for the WLD has been used (

Törnqvist et al., 2007; Nittrouer and Viparelli, 2014b), though that value implicitly assumes that the sandy deposits documented in the WLD stratigraphy (Roberts, 1997; Roberts et al., 2003) are composed entirely of sand. Applying a mean sand content of 70% to the WLD sandy deposits (well above the highest sand content that we have observed in our study area; Table 1) implies a total sand content of 47%, although substantial work will be needed to confirm this. What is clear though is that the sand fraction of the WLD is an order of magnitude greater than of the ACS for a similar input, indicating a much lower

$SRE$. The implication is that $SRE$ varies considerably depending on the specific depositional setting, a finding that we expect applies in most large deltas.

While other researchers have given considerable attention to the importance of coarse-grained sediment as a restoration tool (Nittrouer et al., 2012a; Nittrouer and Viparelli, 2014b), our results highlight the utility of the more plentiful mud load. The goal of river diversions is to maximize their land building potential, which is determined by both sediment supply and $SRE$.

Coarse-grained sediment is largely deposited on the delta front where the $SRE$ tends to be low. As a result, land building with coarse-grained sediment needs a very large sediment input, which implies designing relatively deep diversion channels to extract the more abundant coarse grains deeper in the water column (Meselhe et al., 2012). Consequently, only a few such projects can be operated at any given time and the mud load is mostly lost. Contrary to the viewpoint that only coarse sediment builds land (Nittrouer and Viparelli, 2014b), we find a deltaic feature that is almost entirely fine-grained but still sufficiently

elevated to support agriculture 9 km from the trunk channel, 600 years after its final depositional episode. The high $SRE$ values measured here call into question the use of prograding coastal delta lobes as restoration models.



Although crevasses tend to be shallow and thus extract relatively small amounts of sediment from the trunk channel, their exceptionally high *SRE* results in substantial land accretion. The ACS attained 1−4 cm yr$^{-1}$ accretion rates and built a splay with an area comparable to the WLD even when the discharge was split between Bayou Lafourche and the LMR. Regional analyses of delta-plain topography (Figure 4, and Figure DR2 by Shen et al. (2015)) show that crevasse-splay deposits are the
dominant building blocks of the proximal overbank environment. The abundance of mud-dominated crevasse splays in the MD shows the importance of mud pathways to delta evolution. Their large numbers also make it conceivable that numerous crevasse splays were active at any given time, thus highlighting their potential as land builders compared to a single, terminal delta lobe such as the WLD.

Our results have implications for sediment management strategies in the MD. A significant portion of the vegetated delta plain
is within ~1 m of sea level (Fig. 1a) and will likely submerge by the end of the century if its elevation is not increased (Blum and Roberts, 2009; Coastal Protection and Restoration Authority of Louisiana, 2017). River diversions can exploit high *SRE* values in these environments to maximize gains from the limited sediment load of the modern LMR, and can promote vertical land growth rapidly enough to locally keep up with relative sea-level rise. By contrast, diversions near the open coast are exposed to waves, tides, and currents, reducing their *SRE* and making their long-term viability questionable (Blum and Roberts,
2009). Considering the enormous costs of these projects (Coastal Protection and Restoration Authority of Louisiana, 2017), focusing resources on diversions in emergent settings that are still vegetated is preferable. These issues are not unique to the MD (Ericson et al., 2006; Giosan et al., 2013, 2014; Auerbach et al., 2015), but the political and economic ability to construct system-scale river management infrastructure is not yet present in most other large deltas. We expect this to change as the sea encroaches on major population centers worldwide. Therefore, the lessons learned from such novel attempts to divert sediment
back to the delta plain in the MD have the potential to be impactful globally.

## 5. Conclusions

The late Holocene stratigraphic record of the Mississippi Delta shows that crevasse-splay deposits consist of ~95% mud. The sediment retention efficiency in crevasse splays that form in vegetated environments, sheltered from waves, tides, and currents, exceeds 75% and may well approach 100%. This is dramatically higher than rates observed in prograding coastal delta lobes,
which retain 5 to 30% of the incoming sediment. This contrast highlights the variability in retention rates among different portions of a single delta, and points to the importance of mud pathways for delta evolution. While large volumes of sand are associated with land building in open water, accretion rates of mud alone can be sufficient to locally match relative sea-level rise in a vegetated environment that is isolated from marine processes. Coastal managers can site river diversions to take advantage of locally high sediment retention efficiency by choosing locations that a) are protected from marine processes, b)
contain existing emergent land with established vegetation, and c) are not at risk of imminent submergence. Such sites are likely to be more successful than those on the open coast.



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

**Acknowledgments**

Field assistance by J. Kuykendall, A.G. Nijhuis, M.P. Hijma, Z. Li, and Tulane University Spring 2008 and Fall 2012 Sedimentation and Stratigraphy classes is appreciated. The paper benefited from discussions with M.A. Allison and M.T. Ramirez. This research was supported by US National Science Foundation grant EAR-1148005 to Z.S. and T.E.T. and the Vokes Fellowship to C.R.E., with additional support from the Long-term Estuary Assessment Group program through the Tulane/Xavier Center for Bioenvironmental Research to Z.S., and from the Louisiana Sea Grant Undergraduate Research Opportunities Program to J.M. Schlumberger provided free access to the Petrel software suite. K.M. Straub provided lab equipment used in grain-size analyses.

Author contribution: C.R.E. and Z.S. contributed equally to the paper and should be considered co-first authors. Z.S. and T.E.T. designed the project. Z.S. led all fieldwork. C.R.E. was involved in fieldwork, constructed the 3D model with geostatistical input from C.W., compiled river surveying data, and undertook the suspended sediment modeling. J.M. contributed to fieldwork and conducted and interpreted the grain-size analyses. Z.S. and C.R.E. composed the manuscript with major inputs from T.E.T.

**Competing Interests**

The authors declare that they have no conflict of interest.



**Figures and Tables**

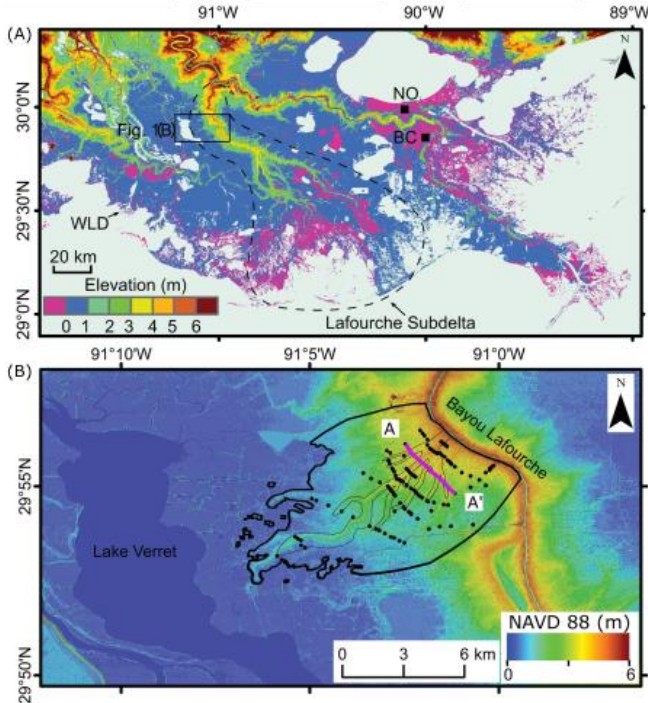

Figure 1. Digital elevation models (DEMs) of the study area. (a) Mississippi Delta, Louisiana, USA. Regional data is derived from the 1/3 arc-second DEM of the US Geological Survey (http://viewer.nationalmap.gov/basic/#startUp), accessed in September, 2015. (b) The Attakapas Crevasse Splay at Napoleonville, Louisiana. Black dots mark core locations. The surface expression of the silty splay deposit used in the 3D model is marked by the thick black line. Thin black lines show alluvial ridges associated with splay channels. The local DEM is derived from Light Detection and Ranging data available from Atlas: The Louisiana State GIS (http://atlas.lsu.edu). NO—New Orleans; BC—Belle Chasse; WLD—Wax Lake Delta; NAVD 88—North American Vertical Datum of 1988.



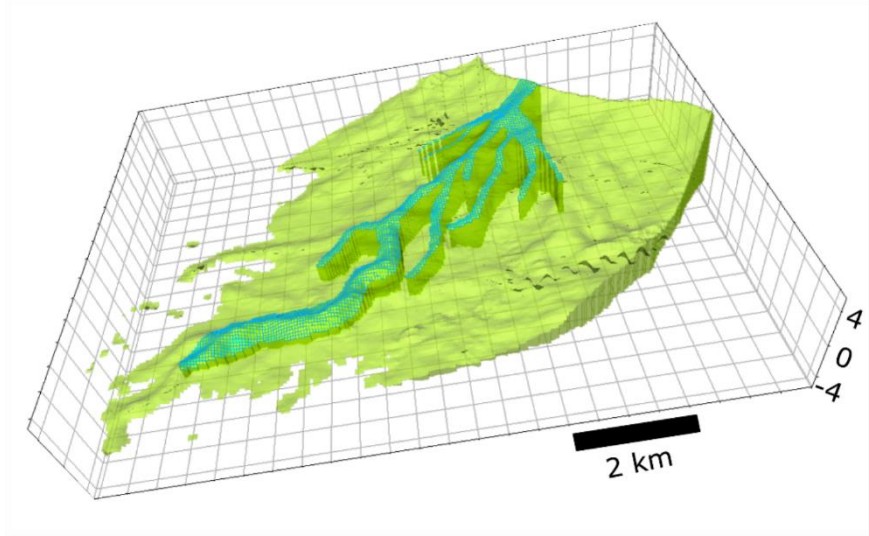

Figure 2. 3D geometry of the Attakapas Crevasse Splay. Vertical axis is in meters relative to NAVD 88. The relatively sandy channel deposits are shown in blue green; light green refers to the silt-dominated portion of the splay. The transparent top bounding surface is the LiDAR-derived modern land surface, and the base is the clay to silt transition shown in Figure 3. Wherever possible, the lateral boundary is chosen to be the intersection of the top and basal bounding surfaces. On the lateral edges, where the deposit does not pinch out and these two surfaces do not meet, the boundary was chosen manually.





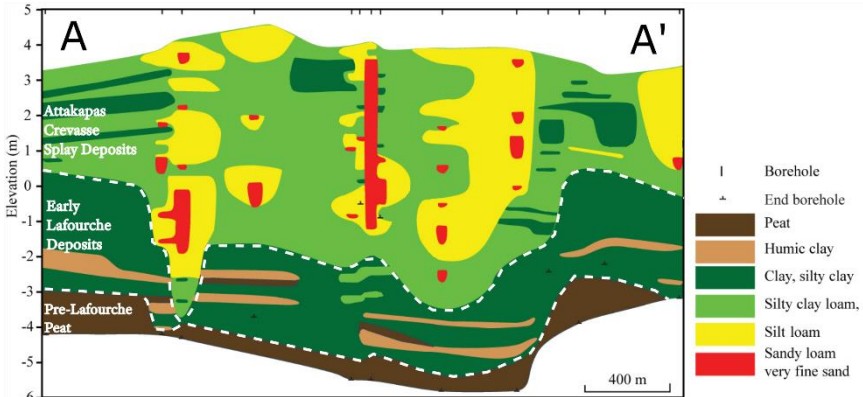

Figure 3: Stratigraphic cross section. The cross section along transect A to A′ (see Figure 1B for location), adapted from Shen et al. (2015). Our 3D model (Figure 2) considers only sediments preserved above the clay to silty clay loam transition. Channel bodies can be seen as deposits of coarser material, often corresponding with alluvial ridges.



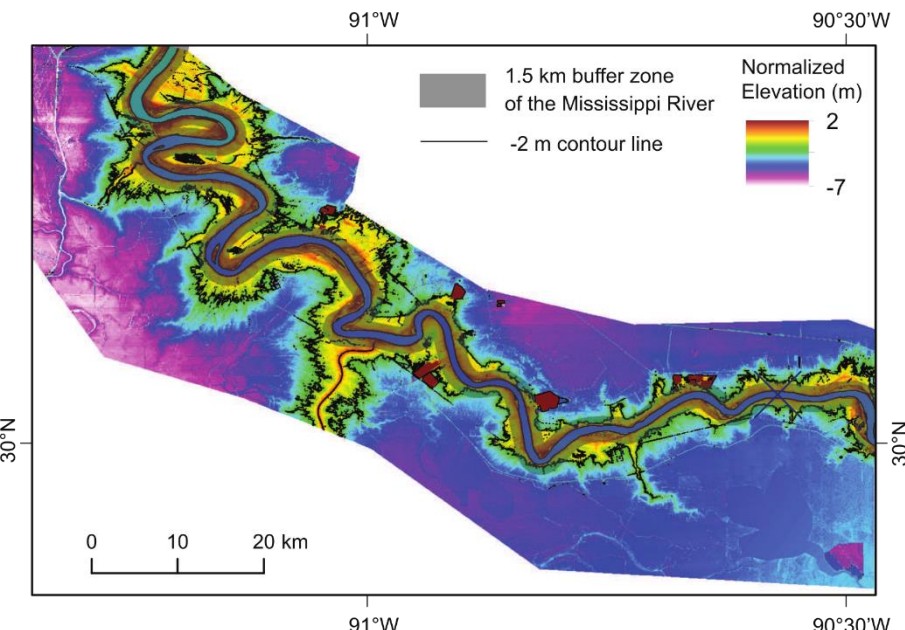

Figure 4. Normalized DEM of the Mississippi River between Baton Rouge and the Bonnet Carre Spillway for crevasse splay identification. The DEM was obtained by subtracting a planar surface that best fits the Mississippi River natural levee long profile from the DEM of the studied reach. A -2 m contour of the normalized DEM and a 1 km buffer zone around the centreline of the Mississippi River are used to identify crevasse splays. The area where the -2 m contour lies outside the buffer zone is identified as formed by a crevasse splay (see Shen et al., 2015, for further explanation), which accounts for ~75% of the bank length in the studied reach. Note the reduction in the extent of crevasse splays downstream of the avulsion site; this is likely due to the fact that the duration of channel activity upstream of this point was considerably longer than in the downstream reach.



| Sediment Texture Class | mean sand fraction | 2σ error | max sand | min sand | n | texture fraction, channel | texture fraction, non-channel |
|---|---|---|---|---|---|---|---|
| **Very Fine Sand (vfS), Sandy Loam (SL)** | 0.2398 | 0.1463 | 0.3159 | 0.1106 | 23 | 0.17 | 0.11 |
| **Silt Loam (SiL)** | 0.0454 | 0.0396 | 0.0662 | 0.0253 | 6 | 0.32 | 0.23 |
| **Silty Clay Loam (SiCL)** | 0.0089 | 0.0294 | 0.0445 | 0.0000 | 10 | 0.40 | 0.46 |
| **Silty Clay (SiC) Clay (C) Humic Clay (HC)** | 0.0066 | 0.0066 | 0.0330 | 0.0000 | 14 | 0.11 | 0.20 |

Table 1. Sediment texture data used to calculate $S_d$. Mean sand fraction, error, and min and max values obtained from 53 sediment samples of the texture classes present in the ACS. Texture fraction in channel/non-channel deposits is calculated from the number of texture descriptions from cores collected inside or outside of our channel boundary, not from interpretations such as that shown in the cross section in Figure 3. Weighting these data by the volumes of channel and non-channel deposits in the splay gives a total splay sand fraction of 0.045. Propagating the 2-σ errors yields a minimum splay sand fraction of 0.028 and a maximum of 0.062.





| | | <15000 | 15000-20000 | 20000-25000 | 25000-30000 | >30000 |
|---|---|---|---|---|---|---|
| River Inputs | bin discharge range ($m^3 s^{-1}$) | <15000 | 15000-20000 | 20000-25000 | 25000-30000 | >30000 |
| | average discharge in bin ($m^3 s^{-1}$) | 9658 | 17417 | 22449 | 27045 | 30959 |
| | days spent at this bin (Oct 1 1989 - Sept 30 2013) | 5116 | 1302 | 1559 | 673 | 116 |
| | total water discharge during record ($m^3 s^{-1}$) | 4.27E+12 | 1.96E+12 | 3.02E+12 | 1.57E+12 | 3.10E+11 |
| | Total sediment discharge during record (metric tons) | 4.98E+06 | 4.28E+06 | 7.06E+06 | 3.69E+06 | 7.13E+05 |
| | average sand load, full channel ($kg s^{-1}$) | 34 | 453 | 965 | 1563 | 2168 |
| | average mud load, full channel ($kg s^{-1}$) | 940 | 2833 | 3563 | 3914 | 3983 |
| | sand fraction of total suspended load (-) | 0.03 | 0.14 | 0.21 | 0.29 | 0.35 |
| *Hydraulic Parameters* | $u*$ estimate ($m s^{-1}$) | 0.06 | 0.06 | 0.06 | 0.09 | 0.09 |
| | z_0 (m) | 0.0005 | 0.0005 | 0.0005 | 0.0005 | 0.0005 |
| Top 5 m | average sand concentration per unit width, top 5 m ($kg m^{-2}$) | 0.00 | 2.60 | 5.53 | 9.05 | 12.55 |
| | average mud concentration per unit width, top 5 m ($kg m^{-2}$) | 21.02 | 63.33 | 79.65 | 58.33 | 59.35 |
| | average sand fraction ($S_i$), top 5 m (-) | 0.00 | 0.04 | 0.06 | 0.13 | 0.17 |

Table 2. Data used to calculate $S_i$. The sand and mud concentrations for each bin are weighted with the respective total water discharge to provide a yearly average sand fraction input to the splay of 0.066.