# Peer review of "Efficient retention of mud drives land building on the Mississippi Delta plain"

_Earth Surface Dynamics, 2017_

## Referee Comment (RC1) · Anonymous Referee #1 · 9 Mar 2017

This manuscript concerns the dynamics of sediment retention in deltaic depositional systems, which is an important area of research within the deltaic community. Restoring drowning deltas, such as the Mississippi River delta, will be a difficult task. One way to accomplish this is using river diversions that nourish areas of drowning with sediment needed to offset sea-level rise. An important part of this idea—and the topic of this manuscript—is how much sediment will be deposited in a diversion. Afterall, the worst case scenario is that all the sand and mud entering the diversion is suspended and transported far away from the desired deposition location. In this way understanding sediment retention is crucial for land building. Presently sediment retention is an ad-hoc factor in delta land building models, and field data are sorely needed to help constrain that value. In these regards I think this manuscript fits in nicely to an important of the larger topic of delta restoration. I also think this manuscript is technically

sound and clearly written. I think the analysis and presentation of SRE calculation for the studied crevasse splay is well done. Indeed, I find it interesting that such a splay contains only 5% sand. I think the present contribution needs to be placed into an appropriate context. Presently the manuscript's main points are 1) a bit overstated given they only studied one crevasse, and 2) lacking a clear context. If these points can be addressed I think the present contribution would be a valuable addition to the scientific literature.

Deltaic restoration studies (such as the one's the authors cite) are focused on coastal depositional features. These coastal features are the first-line of defense against rising relative sea-levels. Because of their position at the coast the sediment and landform are subject to myriad processes, such as river plume deposition, waves, tides, storm surges, etc. All of these processes can keep fine-grained material suspended, enriching the deposit in sand, and moving the mud offshore. THis is probably why features like Wax Lake have such high mud proportions. The splay studied here is located far upstream of the coast and is NOT subjected to the same processes as features like Wax Lake. So in that context, is it surprising that ACS is composed of so much mud? Without some theory, it is hard to say, but my intuition says no. A crevasse splay that does not grow into a standing body of water (like WLD did) might have higher mud retention because the downstream water surface elevation goes to 0 as flood waters recede which forces mud deposition. Coastal features do not have that degree of freedom. In that context, I think the authors can do a better job of placing their crevasse splay into context EARLY in the manuscript. This splay is on a delta plain, but it is hardly a deltaic feature in the same way Wax lake is, yet there are locations where the authors call this splay a deltaic feature. Only after I read the conclusions was it clear to me how the authors viewed their study in the larger context. The early part of the manuscript suffers from a false tension between researchers who view sand as the important land building constituent and the present work. For instance, on lines 12-5 pg. 3 it is true that most studies focus on sand extraction, but it is also true that most of those studies were focusing on coastal features where sand retention is more critical.

Same thing on 28-31 on pg. 10, the viewpoint that sand builds land need to be placed into context. Too often the authors are treating their crevasse splay as a direct comparison to Wax Lake, which is not appropriate given that Wax formed inot a standing body of water and ACS did not. Afterall on line 31 they say their results 'call into question' using Wax Lake as restoration model. I just don't think creating this false tension is helpful. Instead the splay studied here highlights nicely the variability in retention as a function of location of the splay and processes and suggests that one model of SRE is not appropriate. Said another way, this study highlights the need for general theory of what governs SRE in different settings. The authors seemingly take issue with the 'sand-focused' view of land building that has dominated the literature, but I don't think the present contribution does anything to invalidate that previous work. Instead the present contribution shows how complex and varied the problem is—deltas plains consist of shorelines and fluvially dominated zones and SRE at shorelines may apply to other parts of the delta plain.

The title is too grandiose in my opinion. Do the authors really think that efficient retention of mud is what drives land building? They studied only one crevasse splay that was far from the coastline. I would say that the title and some of the primary conclusions need to be revisited and placed in context for the reader. In the same light, I don't see how the present crevasse splay actually is a good model for COASTAL restoration projects. It might be a good model for restoration of the entire delta plain, which consists of coastal and more fluvially dominated zone (like the crevasse studied here). The authors are aware of all these things, given their statements on line 25 pg. 3, but it seems by the end of the paper they forgot that fact. Line 3-5 on pg. 11 seems to suggest that ALL crevasses are created equal. I also find the line on pg. 11 line 15-16 to be odd. Are the authors really suggesting the CPRA focus on emergent settings instead of the coastline? Landloss is MUCH higher near the coast than at the location of this study. Why then focus on emergent settings that are not losing land?

---

## Referee Comment (RC2) · Anonymous Referee #2 · 14 Mar 2017

The authors present an important counterpart to the present literature on delta restoration (primarily the Mississippi) that focuses on sands rather than on the dominant mud fraction.

The demonstration is well done using one example of crevasse splay that the authors generalize to the scale of the whole Mississippi delta, which is dominantly composed of crevasse deposits. One could ask if the crevasse they chose to study is representative and the authors should strengthen their case for this specific point.

Restoration literature uses Wax Lake delta (open coast delta) as a model for crevasse splays, which is of course wrong both in terms of morphodynamics but also in practical terms of retention rates. Contrary to the idea that erosion at the coast is the main mode of land loss in the Mississippi delta, most land is lost on the delta plain and

reconstruction via crevassing would be most effective if appropriate models are used. This study puts things straight, providing such a model and should inspire future efforts of restoration. The authors should underline these better in their conclusions.

**ESurfD**

---

## Referee Comment (RC3) · Anonymous Referee #3 · 14 Apr 2017

This manuscript concerns the important issue of sediment retention, particular that of the finer mud fraction, on floodplains. As set out in the manuscript, the retention of sediment on floodplains is vital for the management of deltaic and wetland envrionments. Furthermore, previous work on this subject has focussed intently on the sand, or coarser, fractions of the sediment load. This work adds a novel and important refocus on the finer fraction which is shown here to represent ∼95% of the sediment volume in the studied crevasse-splay system. As such this manuscript is an important addition to the literature. It is well writen and clearly presented. However, I think the framing of the work, and some of the comparisons to other splay systems needs revisiting, or more justification.

The splay system under consideration in the manuscript is one that developed in a swamp environment (line 27) and which is characteristic of several such features found

along the Mississippi Delta which are key to the maintanence of the landward portions of the MD (line 28 - 30). These splay systems are not developing in open water basins at the marine edges of the delta where they are being effected by tides and waves in the Gulf of Mexico, rather they likely develop in empty basins where the main control is topography and fluvial inputs and where mud is potentially rapidly deposited (rather than resuspended) as sediment enters the basin. It is therefore interesting that the main area of comparison is with the Wax Lake Delta, which is a coastal deltaic deposit developing at the marine interface and is influenced by tides and waves. It seems to me that these two systems are not directly comparable as the marine influences are likely to effect the processes of deposition and resuspension which occur at the splay edges, thus impacting the proportions and locations of the sediment fractions understudy. It would be more useful to compare the findings of this work to other terrestrial/fluvial crevasse-splay systems found along the main channels of the "inland" MD rather than deltaic deposits at the marine interface.

It would also be interesting to have a discussion around the errors in the authors estimates of ratings curves used to estimate sediment load and fractionations, and also the water levels in the assumed trunk channels from which the splay eminates. Is there any hysteresis displayed in the sediment ratings curves that could impact upon the functioning of the splays? How good is the fit of the ratings curves and what the propograted errors through the estimates of sediment concentration and dischagre? The estimates of SRE the authors report are likely to vary with these and it would be useful to have an idea of the sensitivity of the metrics used by the authors to characterise the ACS to these input parameters.

---

## Author Comment (AC1) · 14 Apr 2017

**Response to Anonymous Reviewer 1 on* "Efficient retention of mud drives land building on the Mississippi Delta plain" *by* C. Esposito et al.**

**Christopher R. Esposito** *on behalf of* **Zhixiong Shen, Torbjörn E. Törnqvist, Jonathan Marshak, Christopher White**

*We appreciate the constructive reviews submitted by Anonymous Referees 1 and 2 (AR1 and AR2). We provide responses to the AR1's comments below, in underlined italics.*

This manuscript concerns the dynamics of sediment retention in deltaic depositional systems, which is an important area of research within the deltaic community. Restoring drowning deltas, such as the Mississippi River delta, will be a difficult task. One way to accomplish this is using river diversions that nourish areas of drowning with sediment needed to offset sea-level rise. An important part of this idea, and idea and the topic of this manuscript is how much sediment will be deposited in a diversion. Afterall, the worst case scenario is that all the sand and mud entering the diversion is suspended and transported far away from the desired deposition location. In this way understanding sediment retention is crucial for land building. Presently sediment retention is an ad-hoc factor in delta land building models, and field data are sorely needed to help constrain that value. In these regards I think this manuscript fits in nicely to an important of the larger topic of delta restoration. I also think this manuscript is technically sound and clearly written. I think the analysis and presentation of SRE calculation for the studied crevasse splay is well done. Indeed, I find it interesting that such a splay contains only 5% sand. I think the present contribution needs to be placed into an appropriate context. Presently the manuscript's main points are 1) a bit overstated given they only studied one crevasse, and 2) lacking a clear context. If these points can be addressed I think the present contribution would be a valuable addition to the scientific literature.

*A strong case can be made that the results obtained from our study area, the Attakapas Crevasse Splay (ACS), are applicable broadly to the Holocene Mississippi Delta plain. First, we presented regional topographic analyses of the Mississippi Delta that show crevasse splay features as the primary building blocks of the proximal overbank environment surrounding the major distributary channels. We have rearranged our figures so that this information (Figure 2, formerly Figure 4) is now featured more prominently in the Introduction, and will be in the forefront of the reader's mind. Second, we added references in the Discussion (Section 4.2) to McFarlan (1961) and Frazier (1967), who studied the stratigraphy of the Mississippi Delta on a*

*much larger scale, to strengthen the case that our results are in line with other characterizations of proximal overbank deposits in the Mississippi Delta.*

*These changes also address AR2's request that we need to strengthen the case that our study site is representative of a wider region.*

Deltaic restoration studies (such as the one's the authors cite) are focused on coastal depositional features. These coastal features are the first-line of defense against rising relative sea-levels. Because of their position at the coast the sediment and landform are subject to myriad processes, such as river plume deposition, waves, tides, storm surges, etc. All of these processes can keep fine-grained material suspended, enriching the deposit in sand, and moving the mud offshore. THis is probably why features like Wax Lake have such high mud proportions.

*The mechanism for loss of fine grained material described above ("waves, tides, storm surges, etc.") is what we referred to generally as "marine processes on the open coast" at line 22 on page 3. We have changed this language to be more specific in the revised manuscript.*

The splay studied here is located far upstream of the coast and is NOT subjected to the same processes as features like Wax Lake. So in that context, is it surprising that ACS is composed of so much mud?

*We also do not find it surprising that the ACS is composed of so much more mud than the Wax Lake Delta. However, because the Wax Lake Delta has been the focus of so much of the literature on coastal restoration in deltas, we feel that it is important to document that its sediment retention efficiency (SRE) is low compared to other deltaic environments.*

Without some theory, it is hard to say, but my intuition says no. A crevasse splay that does not grow into a standing body of water (like WLD did) might have higher mud retention because the downstream water surface elevation goes to 0 as flood waters recede which forces mud deposition. Coastal features do not have that degree of freedom. In that context, I think the authors can do a better job of placing their crevasse splay into context EARLY in the manuscript.

*As above, we are not surprised that the ACS has a higher SRE than the WLD, but our objective in comparing the two is to quantify the difference, something that to our knowledge hasn't been done before. We stated early on (in the Abstract and in the Introduction) that we are comparing SRE in the ACS to that of the WLD. To that end, we actually have reasons to believe that the crevasse splay that we study represents, if anything, an upper limit on sand content (and therefore a lower limit on SRE) of crevasse splays in the Mississippi Delta. We have consolidated elements of Section 4.2 to make this point more clearly. We now refer to the fact that crevasse splays are fundamental delta building blocks (as shown by Figure 2) in the same paragraph where we explain why the ACS is likely to be sandier than surrounding splays. We have also added text to Section 2.2 to clarify that our choices are conservative ones in the context of coastal restoration planning. This change also addresses AR2's request that we strengthen the case that our study site is representative.*

This splay is on a delta plain, but it is hardly a deltaic feature in the same way Wax lake is, yet there are locations where the authors call this splay a deltaic feature.

*We find this comment confusing. If the splay is on the delta plain, it counts as a "deltaic feature". This is consistent with how deltas are defined in widely used textbooks in coastal geomorphology.*

Only after I read the conclusions was it clear to me how the authors viewed their study in the larger context. The early part of the manuscript suffers from a false tension between researchers who view sand as the important land building constituent and the present work. For instance, on lines 12-5 pg. 3 it is true that most studies focus on sand extraction, but it is also true that most of those studies were focusing on coastal features where sand retention is more critical. Same thing on 28-31 on pg. 10, the viewpoint that sand builds land need to be placed into context.

*We do not agree that this tension is false. Some of the literature that we cite focuses on the river-side impacts of diversion where sand is clearly important. However, there is also a line of thinking that sand is critically important to land building in a restoration context. To take one example, we cite Nittrouer and Viparelli's 2014 paper in Nature Geoscience, titled "Sand as a stable and sustainable resource for nourishing the Mississippi River delta", which includes the following text in the abstract:*

> *"...sand – which accounts for ~50-70% of modern and ancient Mississippi delta deposits but comprises ~20% of the sampled portion of the total load – could be more important than mud for subaerial delta growth."*

*Our data show that this is almost certainly not the case, and that substantial land building can be obtained with mud alone. We agree that our work highlights the variability of SRE in a delta, and the need for a general theory of what governs SRE in different settings. In fact, we see our comparison with the Wax Lake Delta as important from this perspective. But we also see a very real tension about what is a reasonable value for delta-averaged SRE that must be resolved.*

Too often the authors are treating their crevasse splay as a direct comparison to Wax Lake, which is not appropriate given that Wax formed inot a standing body of water and ACS did not. Afterall on line 31 they say their results 'call into question' using Wax Lake as restoration model. I just don't think creating this false tension is helpful.

*Again, we deliberately compare the SRE in the ACS to that of the Wax Lake Delta, and we find that the SRE of the ACS was much higher. Whatever the reason for this, it suggests that restoration projects that result in environments like the Wax Lake Delta will waste limited sediment resources.*

Instead the splay studied here highlights nicely the variability in retention as a function of location of the splay and processes and suggests that one model of SRE is not appropriate. Said another way, this study highlights the need for general theory of what governs SRE in different settings.

*We agree with this comment, and in fact make this very point in the second paragraph of the Introduction, and again in the second paragraph of Section 4.2.*

The authors seemingly take issue with the 'sand-focused' view of land building that has dominated the literature, but I don't think the present contribution does anything to invalidate that previous work. Instead the present contribution shows how complex and varied the problem isâ˘Aˇ Tdeltas plains consist of shorelines and fluvially dominated zones and SRE at shorelines may apply to other parts of the delta plain.

*As stated earlier, we agree that one of our contributions is to highlight the variability of SRE. More importantly, it shows that a principal building block of the Mississippi Delta, crevasse splays, are primarily formed by mud. This latter finding is important in showing that mud is much more efficient in land building than previously thought and should play a critical role in coastal restoration given its ~80% contribution to the Mississippi River sediment load.*

The title is too grandiose in my opinion.

*We address this comment below.*

Do the authors really think that efficient retention of mud is what drives land building? They studied only one crevasse splay that was far from the coastline. I would say that the title and some of the primary conclusions need to be revisited and placed in context for the reader. In the same light, I don't see how the present crevasse splay actually is a good model for COASTAL restoration projects. It might be a good model for restoration of the entire delta plain, which consists of coastal and more fluvially dominated zone (like the crevasse studied here).

*This comment seems to stem from a disagreement over what constitutes "coastal restoration". We note that "coastal restoration" does not refer only to activity at the shorefront. For example, the Louisiana Coastal Zone (http://www.dnr.louisiana.gov/assets/OCM/CoastalZoneBoundary/CZB2012/CZB_ReviewHando uts.pdf), which hosts some of the largest coastal restoration initiatives on earth, extends inland hundreds of kilometers from the shoreline. It is defined on the basis of elevation, vegetation type, and – critically – susceptibility to inundation by storm surge. This definition is in line with that of textbooks and classic literature (e.g. Dalrymple et al., 1992; M. R Leeder, 1999), where the inland limit of tidal influence is often used to mark the edge of the coastal zone. Thus shores, beaches, barrier islands, prograding deltas, and the relatively inland fluvio-deltaic settings like the one we studied are all coastal environments.*

The authors are aware of all these things, given their statements on line 25 pg. 3, but it seems by the end of the paper they forgot that fact.

*We do not believe that a careful reader will think that we "forgot" that our results do not apply to the shoreline. The contrast between deposition in a crevasse splay and deposition at the shorefront is explicitly discussed in the Abstract, Introduction, Discussion, and Conclusion.*

Line 3-5 on pg. 11 seems to suggest that ALL crevasses are created equal.

*We do not believe that all crevasses are created equal. As mentioned above, we have rearranged sections 4.2 and 2.2 to make it more clear that our results from the ACS likely show an upper limit for sand content in a crevasse splay, and consequently provide a lower limit for SRE.*

I also find the line on pg. 11 line 15-16 to be odd. Are the authors really suggesting the CPRA focus on emergent settings instead of the coastline?

*Yes, we are really suggesting that the CPRA (Coastal Protection and Restoration Agency of Louisiana) should focus on emergent settings instead of the coastline. There is widespread understanding that the current shoreline of the Mississippi Delta is not sustainable with projected rates of relative sea level rise (see, for example, Oppenheimer and Alley, (2016) ), that significant drowning is inevitable, and that managing the coastal zone will involve painful choices of which areas to prioritize. There are many cases where diversion to an inland emergent setting will allow it to keep pace with sea level when it otherwise would have been drowned, and other cases where even aggressive action near the shoreline will only waste valuable sediment resources. In keeping with these observations, the Louisiana Master Plan includes a "land maintained" category in their predictions to indicate areas that would have been changed from land to water, but were prevented from doing so by a coastal restoration project. We agree with this approach.*

*AR1 suggests that our title is not accurate because crevasse splays do not necessarily change open water into land. But the implicit knowledge driving the above discussion about diversion location is that building land elevation is more important in a restoration context than building land extent. We therefore feel that our current title is appropriate.*

Landloss is MUCH higher near the coast than at the location of this study. Why then focus on emergent settings that are not losing land?

*We are not proposing that diversions be used in the vicinity of our study site. AR1 is correct that land loss is not very high in that area. But as AR2 points out, "Contrary to the idea that erosion at the coast is the main mode of land loss in the Mississippi delta, most land is lost on the delta*

*plain and reconstruction via crevassing would be most effective if appropriate models are used."
We agree with this statement, and feel that we have presented our study site as a possible
restoration model, not as a location in need of restoration.*

*REFERENCES*

*Dalrymple, R. W., Zaitlin, B. A. and Boyd, R.: Estuarine facies models; conceptual basis and
stratigraphic implications, J. Sediment. Res., 62(6), 1130–1146, 1992.*

*Frazier, D. E.: Recent Deltaic Deposits of the Mississippi River: Their Development and
Chronology, Gulf Coast Assoc Geol Soc Trans., 17, 287–315, 1967.*

*M. R Leeder: Sedimentology and sedimentary basins: from turbulence to tectonics, Blackwell
Science, Oxford ; Malden, MA., 1999.*

*McFarlan, E.: Radiocarbon Dating of Late Quaternary Deposits, South Louisiana, Geol. Soc.
Am. Bull., 72(1), 129–158, doi:10.1130/0016-7606(1961)72[129:RDOLQD]2.0.CO;2, 1961.*

*Nittrouer, J. A. and Viparelli, E.: Sand as a stable and sustainable resource for nourishing the
Mississippi River delta, Nat. Geosci., 7(5), 350–354, doi:10.1038/ngeo2142, 2014.*

*Oppenheimer, M. and Alley, R. B.: How high will the seas rise?, Science, 354(6318), 1375–1377,
doi:10.1126/science.aak9460, 2016.*

---

## Author Comment (AC3) · 18 Apr 2017

**Response to Anonymous Reviewer 3 on* "Efficient retention of mud drives land building on the Mississippi Delta plain" *by* C. Esposito et al.**

**Christopher R. Esposito** o*n behalf of* **Zhixiong Shen, Torbjörn E. Törnqvist, Jonathan Marshak, Christopher White**

*We appreciate the constructive reviews submitted by Anonymous Referees 1, 2 and 3 (AR1, AR2, and AR3). We provide responses to the AR3's comments below, in underlined italics.*

This manuscript concerns the important issue of sediment retention, particular that of the finer mud fraction, on floodplains. As set out in the manuscript, the retention of sediment on floodplains is vital for the management of deltaic and wetland envriaments. Furthermore, previous work on this subject has focussed intently on the sand, or coarser, fractions of the sediment load. This work adds a novel and important refocus on the finer fraction which is shown here to represent ~95% of the sediment volume in the studied crevasse-splay system. As such this manuscript is an important addition to the literature. It is well writen and clearly presented. However, I think the framing of the work, and some of the comparisons to other splay systems needs revisiting, or more justification.

The splay system under consideration in the manuscript is one that developed in a swamp environment (line 27) and which is characteristic of several such features found along the Mississippi Delta which are key to the maintanence of the landward portions of the MD (line 28 - 30). These splay systems are not developing in open water basins at the marine edges of the delta where they are being effected by tides and waves in the Gulf of Mexico, rather they likely develop in empty basins where the main control is topography and fluvial inputs and where mud is potentially rapidly deposited (rather than resuspended) as sediment enters the basin. It is therefore interesting that the main area of comparison is with the Wax Lake Delta, which is a coastal deltaic deposit developing at the marine interface and is influenced by tides and waves. It seems to me that these two systems are not directly comparable as the marine influences are likely to effect the processes of deposition and resuspension which occur at the splay edges, thus impacting the proportions and locations of the sediment fractions understudy.

*This comment echoes concerns expressed by AR1. We do not view the ACS and the WLD as "comparable" in the sense that they should be expected to behave similarly. Rather, we compare the two to emphasize the variability in SRE between different environments on a delta, and point out the value in understanding that variability to coastal restoration efforts. We have made changes to Sections 4.2 and 2.2 of our manuscript to make sure that this intent is more clear.*

It would be more useful to compare the findings of this work to other terrestrial/fluvial crevasse-splay systems found along the main channels of the "inland" MD rather than deltaic deposits at the marine interface.

*We have reworked sections our manuscript (Sections 4.2 and 2.2)  to emphasize that the ACS is likely to represent an upper limit on sand content for crevasse splays in the region, and therefore a lower limit on SRE. And we have reordered our figures so that the figure showing that crevasse splays are fundamental building blocks of the delta plain (Figure 2, Formerly Figure 4) appears early in the manuscript.*

It would also be interesting to have a discussion around the errors in the authors estimates of ratings curves used to estimate sediment load and fractionations, and also the water levels in the assumed trunk channels from which the splay eminates. Is there any hysteresis displayed in the sediment ratings curves that could impact upon the functioning of the splays? How good is the fit of the ratings curves and what the propogated errors through the estimates of sediment concentration and dischagre? The estimates of SRE the authors report are likely to vary with these and it would be useful to have an idea of the sensitivity of the metrics used by the authors to characterise the ACS to these input parameters.

*We have expanded the last paragraph of Section 4.1, which explains our error analysis and sensitivity testing. All of these calculations are present in the supplemental spreadsheet.*

*We also added a sentence in the second paragraph of Section 2.3 to indicate that we binned the data so that sediment load hysteresis is not likely to be an issue.*

---

## Author Comment (AC2)

**Response to Anonymous Reviewer 2 on* "Efficient retention of mud drives land building on the Mississippi Delta plain" *by* C. Esposito et al.**

**Christopher R. Esposito** o*n behalf of* **Zhixiong Shen, Torbjörn E. Törnqvist, Jonathan Marshak, Christopher White**

*We appreciate the constructive reviews submitted by Anonymous Referees 1 and 2 (AR1 and AR2). We provide point-to-point responses to the AR2's comments below, in underlined italics.*

The authors present an important counterpart to the present literature on delta restoration (primarily the Mississippi) that focuses on sands rather than on the dominant mud fraction. The demonstration is well done using one example of crevasse splay that the author generalize to the scale of the whole Mississippi delta, which is dominantly composed of crevasse deposits. One could ask if the crevasse they chose to study is representative and the authors should strengthen their case for this specific point.

*We have rearranged our figures so that Figure 2 (formerly Figure 4) is now featured more prominently in the introduction. This figure presents a regional topographic analysis of the Mississippi Delta to show that crevasse splay features are primary building blocks of the proximal overbank environment surrounding major distributary channels. We have also consolidated elements in the discussion (Section 4.2) to make it clear that the Attakapas Crevasse Splay represents, if anything, an upper limit on sand content (and therefore a lower limit on SRE). Furthermore, we have added references to classic literature (McFarlan, 1961; Frazier, 1967) to strengthen the point that our mud dominated study site is representative of proximal overbank deposits in the Mississippi Delta. These changes also address concerns by AR1.*

Restoration literature uses Wax Lake delta (open coast delta) as a model for crevasse splays, which is of course wrong both in terms of morphodynamics but also in practical terms of retention rates. Contrary to the idea that erosion at the coast is the main mode of land loss in the Mississippi delta, most land is lost on the delta plain and reconstruction via crevassing would be most effective if appropriate models are used. This study puts things straight, providing such a model and should inspire future efforts of restoration. The authors should underline these better in their conclusions.

*We have adjusted our conclusion to include language emphasizing the importance of appropriate depositional models in restoration efforts.*

---

## Author Response (AR2)

**Response to Editor D.R Parsons on "Efficient retention of mud drives land building on the Mississippi Delta plain" by C. Esposito et al.**

**Christopher R. Esposito o*n behalf of* Zhixiong Shen, Torbjörn E. Törnqvist, Jonathan Marshak,**

5 **Christopher White**

Editor D.R Parsons comments:

10 "The authors have addressed many of the main points raised in review via some reorganization
and adding some clarity to the text in places. My major concern was that
two of the reviewers questioned the approach of comparing the prototype to Wax Lake
Delta...the authors rebut this and highlight what their aims were in such a comparison.
In my view this is not yet clear enough and I would encourage them to look again at the
15 pitch of the introduction in this regard. I am happy to recommend that the paper title
remains as is as long as this approach is more fully articulated in the introduction - if
two expert reviewers have the same view and have read the same (wrong) intent this
warrants a closer look from the reviewers - the reorganisation goes some way towards
this but does not fully address the concerns. This apart the paper reads well and I
20 recommend publication - subject to the above. Daniel R. Parsons"

*We appreciate the opportunity to publish our work in Earth Surface Dynamics. We have addressed the Editor's*
25 *concerns by rewriting the second paragraph of the Introduction. Our marked edits are shown below.*

[revised manuscript text omitted]